# Feeling Anxious amid the COVID-19 Pandemic: Psychosocial Correlates of Anxiety Symptoms among Filipina Domestic Helpers in Hong Kong

**DOI:** 10.3390/ijerph17218102

**Published:** 2020-11-03

**Authors:** Nelson C. Y. Yeung, Bishan Huang, Christine Y. K. Lau, Joseph T. F. Lau

**Affiliations:** The Jockey Club School of Public Health and Primary Care, The Chinese University of Hong Kong, Hong Kong, China; charlottebh@cuhk.edu.hk (B.H.); ykclau@cuhk.edu.hk (C.Y.K.L.); jlau@cuhk.edu.hk (J.T.F.L.)

**Keywords:** anxiety, COVID-19, domestic helpers, Filipina, workload, worries

## Abstract

The COVID-19 pandemic negatively impacts psychological well-being (e.g., anxiety symptoms) among the general population of Hong Kong and migrant Filipina domestic helpers (FDHs). Having to live with the employers by law, FDHs’ working environment might affect their well-being during COVID-19 (e.g., household crowdedness/size, insufficiency of protective equipment against COVID-19, increased workload). Research has suggested that coping resources (e.g., social support, COVID-19-related information literacy) and COVID-19-specific worries are associated with people’s well-being during COVID-19. This study examined the psychosocial correlates of probable anxiety among FDHs in Hong Kong amid the COVID-19 pandemic. By purposive sampling, FDHs (*n* = 295) were recruited and invited to complete a cross-sectional survey. Participants’ working environment (crowdedness, household size), COVID-19 job arrangements (workload, provision of protective equipment), coping resources (social support, COVID-19 information literacy), COVID-19-specific worries (contracting COVID-19, getting fired if contracting COVID-19), and anxiety symptoms were measured. Multivariate regression results showed that the insufficiency of protective equipment (OR = 1.58, 95%CI: 1.18, 2.11), increased workload (OR = 1.51, 95%CI: 1.02, 2.25), and worries about being fired if getting COVID-19 (OR = 1.32, 95%CI: 1.04, 1.68) were significantly associated with probable anxiety. This was one of the earliest studies to indicate that job arrangements and COVID-19-specific worries significantly contributed to FDHs’ anxiety symptoms. Our findings shed light on the importance of addressing employment-related rights and pandemic-specific worries through interventions among FDHs in Hong Kong during pandemic situations.

## 1. Introduction

The coronavirus pandemic (COVID-19) has become an international public health emergency, posing continuous threats to lives and healthcare systems worldwide. As of 26 October 2020, there have been 42,745,212 confirmed COVID-19 cases from more than 200 countries/regions [1]. Since the first reported case on 23 January 2020, Hong Kong has reported 5,304 COVID-19 cases and 105 deaths as of 26 October 2020 [2]. Similarly to other countries/regions, the Hong Kong government has implemented multiple preventive measures, including cross-border travel restrictions, regulations for social distancing, and quarantine arrangements, to slow down the spread of the virus [2]. However, studies have found that COVID-19 and its relevant control measures bring enormous psychological impacts on the general populations in different countries (e.g., stress, depression, and anxiety) [3]. For example, a recent study found that 14% and 25.4% of the general population in Hong Kong reported a moderate level of anxiety symptoms and deteriorated mental health since the pandemic [4]. The COVID-19 pandemic is, therefore, a highly stressful experience for the Hong Kong general population.

### 1.1. Domestic Helpers’ Mental Health during the COVID-19 Pandemic

The COVID-19 pandemic not only affects the general population of Hong Kong, but also its migrant workers. Hong Kong has been one of the most popular destinations for Filipina domestic helpers (FDHs) to seek employment. In 2018, the number of FDHs employed in Hong Kong was 386,075, with a majority coming from the Philippines [5]. During the COVID-19 pandemic, it was common to see FDHs’ workload increased as they were responsible for cleaning the households and taking care of their employers’ families when many public facilities were closed [6]. Many of them reported difficulties in obtaining protective equipment for themselves, such as masks from the pharmacy [6]. Not only worrying about personal health, FDHs also expressed worries about losing their job in the pandemic times [6]. Therefore, FDHs in Hong Kong are subject to multiple sources of stressors in the pandemic (e.g., increased workload, having insufficient protective equipment, pandemic-related worries). To deal with stressors related to the COVID-19 pandemic, people also tend to draw on coping resources. The Stress and Coping model [7] postulates that people’s utilization of available coping resources (e.g., social support, pandemic-related information) is associated with their well-being during stressful times. This postulation has been supported by recent studies in the context of COVID-19 [8,9].

During the pandemic, FDHs’ opportunities to seek emotional and material support from their peers might have been reduced due to restrictions on public gatherings during their weekly day off. On the other hand, some FDHs found it difficult to understand the pandemic-related information released in the media, so they might not have been fully informed about the evolution of the pandemic [10]. All of the above suggested that the needs of FDHs in Hong Kong might be largely neglected [11,12]. Understanding the factors associated with FDHs’ adjustments to the COVID-19 pandemic is, therefore, important. Anecdotal evidence has also suggested that such conditions may also put the FDHs at higher risks for anxiety symptoms (e.g., restlessness, trouble relaxing, nervousness) [13,14]. Therefore, this study aimed to examine how job settings and arrangements (e.g., workload, sufficiency of protective equipment), coping resources (social support, COVID-19 information literacy), and COVID-19-specific worries might contribute to anxiety symptoms among FDHs in Hong Kong.

### 1.2. Working Household Environment and Job Arrangements might Contribute to FDHs’ Anxiety Symptoms during COVID-19

Given that COVID-19 is highly infectious, the working household environment could be particularly important to FDHs’ well-being. Restricted by the immigration live-in law [15], FDHs in Hong Kong could not live outside of their employers’ household. As there is generally little living space in Hong Kong households, it is common for FDHs to be forced to live and sleep in a crowded and unsatisfactory environment. A prior study found that the level of satisfaction towards the working household environment was associated with mental well-being among FDHs in Hong Kong [16]. Recent research has also suggested that the chance of having COVID-19 infection might be higher in families with larger household sizes (i.e., more people living in the same household) [17] and greater household crowding [18]. Those household environmental factors could be associated with risks for infection. We therefore expected that household environment variables (household crowdedness and household size) might contribute to anxiety symptoms among FDHs amid the COVID-19 pandemic.

Job arrangements during the COVID-19 pandemic may also play a role in FDHs’ well-being. Given that there are no legal regulations on maximum working hours for FDHs in Hong Kong [19], FDHs are vulnerable to exploitation (e.g., having to work extra hours without compensation). A recent survey indicated that many FDHs have reported increased workload during the COVID-19 pandemic in Hong Kong (e.g., unreasonable amounts of cleaning work, household chores, and queuing for a long time to buy daily essentials for the employers’ household) [20,21]. Even outside of pandemic situations, work has been identified as the primary source of stress among FDHs in Singapore [22]; higher working hours were associated with lower quality of life among FDHs in Hong Kong [23]. Furthermore, anecdotal evidence suggests that many FDHs cannot obtain protective equipment for themselves, such as masks from the pharmacy, as their employers fail to provide them with adequate equipment [10]. The insufficiency of protective equipment might also heighten the risks for mental distress among FDHs. Therefore, we expected that increased workload during COVID-19 and insufficiency of protective equipment would be associated with more anxiety symptoms among FDHs in Hong Kong.

### 1.3. Social Support, COVID-19 Information Literacy, and Anxiety Symptoms

In response to COVID-19-related stressors, interpersonal and information resources matter. Social support refers to individuals’ support from others in the social system for coping with stressors [24]. It has been found that FDHs’ opportunities to seek emotional and material support from their peers might have been reduced due to the COVID-19 pandemic and social distancing measures [20]; the role of social support is, therefore, important. Previously, the protective role of social support in psychological well-being was found among Filipina domestic helpers in Hong Kong [25]. We expected that social support would be associated with lower anxiety symptoms among FDHs coping with the COVID-19 pandemic.

Furthermore, understanding of trauma-related information may count. When confronted with a highly stressful event (e.g., a disease pandemic), individuals need to be able to make sense of the current event-related information to facilitate better adjustments to the events [26,27]. Potentially due to the lack of reliable access to timely COVID-19 information, anecdotal evidence has suggested that FDHs found it difficult to understand the evolution of the pandemic and appropriate ways to protect themselves [10]. Scholars also suggest that literacy in COVID-19 information might help people to grasp the reasons behind the health recommendations, guide their actions, and reflect on their adjustment process through the pandemic [28]. This may be particularly important in the COVID-19 context, as the pandemic is still evolving without a clear endpoint yet. A recent study also found that timely and accurate information was negatively associated with anxiety symptoms among the general population in China [29]. We therefore expected that higher COVID-19 information literacy would be associated with lower anxiety symptoms among FDHs.

### 1.4. COVID-19-Specific Worries and Anxiety Symptoms

COVID-19-specific worries could also affect to people’s mental well-being during the pandemic. For example, worries about the consequences of getting COVID-19 (e.g., maintaining employment during the subsequent economic downturn, getting treated for COVID-19 if contracting it) were associated with anxiety symptoms among the general population in China [30] and young adults in the US [31]. Anecdotal evidence has suggested that FDHs in Hong Kong are subject to health-related worries about contracting COVID-19, job uncertainties, and pressure to financially support families in their home country during the pandemic [32]. We expected that higher levels of COVID-19-related worries would be positively associated with FDHs’ anxiety symptoms.

### 1.5. Purpose and Hypotheses

This study examined the psychosocial correlates of anxiety symptoms among FDHs in Hong Kong amid the COVID-19 pandemic. Based on the literature reviewed, we hypothesized that work settings, job arrangements during COVID-19, and worries about COVID-19 would contribute significantly to anxiety symptoms. Specifically, higher levels of household crowdedness, household size, insufficiency of protective equipment, workload during COVID-19, and COVID-19-specific worries would be associated with higher anxiety symptoms. In contrast, higher levels of social support and COVID-19 information literacy would be associated with lower anxiety symptoms.

## 2. Methods

### 2.1. Design and Recruitment Strategies

This study used a cross-sectional design with questionnaires as the method of data collection. Inclusion criteria for the participants were: (1) Filipina domestic helpers aged over 18, (2) who had been working for at least three months in Hong Kong, and (3) were able to answer the questions of the questionnaire in English. In Hong Kong, 98% of foreign domestic helpers are female [5]. With purposive sampling, 295 female domestic helpers were recruited from popular gathering venues in Hong Kong Island, Kowloon, and the New Territories (i.e., Hong Kong’s three major territories). It is common for FDHs to gather at public venues like parks and playgrounds during weekends [33]. Similar methods have been used in previous studies on FDHs in Hong Kong [16,25]. Prior to participation, prospective participants were briefed about the study during recruitment on-site. After giving informed consent, they were asked to complete a self-administered paper-and-pencil questionnaire. Upon completing the questionnaire, participants received a supermarket voucher (worth 50 HKD) for compensation of their time. The study was conducted on 9 May and 17 May 2020, just after the government’s loosening of social gathering restrictions in public places from four to a maximum of eight people [34]. The study protocol was approved by the Survey and Behavioral Research Ethics Committee at the authors’ institution (Protocol no. SBRE-19-623).

### 2.2. Measures

#### 2.2.1. Anxiety Symptoms

The General Anxiety Disorder-7 (GAD-7) was used to measure participants’ levels of anxiety over the previous two weeks [35]. On a four-point Likert scale (0 as not at all to 3 as nearly every day), a higher sum score from all items (e.g., “feeling nervous, anxious, or on edge”) indicated more frequent anxiety symptoms. The GAD-7 has been shown to be reliable and valid in community populations [3], with a cut-off point of ≥10 indicating a moderate level of anxiety [35]. Regarding validity, the scale was associated with depressive symptoms and poorer self-rated mental health [36]. Cronbach’s alpha for this sample was 0.91.

#### 2.2.2. Working-Environment-Related Variables

Participants’ working environment was examined using two items. Similarly to other studies examining the role of household environment in mental health outcomes in the context of COVID-19 [29], participants were asked to report the number of people living in the working household, as well as to rate the level of crowdedness in their working household (on a six-point scale; 1 as very spacious, 6 as very crowded).

#### 2.2.3. Workload during COVID-19

The three-item Effort subscale from the Effort–Reward Imbalance Scale [37] was used to assess the participants’ workload during the COVID-19 pandemic (e.g., over the past months, my job has become more and more demanding). On a four-point Likert scale (1 as strongly disagree, 4 as strongly agree), a higher mean score from the item responses represented a higher level of workload during COVID-19. In a migrant worker context, its English version was also found to be reliable and valid among migrant hotel room cleaners (including Filipino) in the US [38]. Other studies also found that the scale was reliable and valid among employees in China [39] and migrant factory workers in China [40]. The scale was associated with poorer mental health [37,39]. Cronbach’s alpha was 0.87 in this study.

#### 2.2.4. Insufficiency of Protective Equipment

Three items were specifically developed to measure participants’ insufficiency of protective equipment against COVID-19 (i.e., “I made my own face masks”, “My employer did not provide me with any protective face masks or hand sanitizers”, and “I needed to reuse my face masks”). On a five-point scale (1 as not at all true, 5 as always true), a higher mean score from the item responses represented a higher insufficiency of protective equipment. Cronbach’s alpha was 0.71.

#### 2.2.5. Social Support

Eight items were developed based on the Cancer Perceived Agents of Social Support (CPASS) [41], which measured the participants’ receipt of emotional and material support from multiple resources during the COVID-19 pandemic. On a five-point Likert scale (1 as not at all, 5 as very much), participants were asked to indicate the extent to which they felt that they received material/emotional support from different sources, including their employers, family members, friends, and community organizations. A sample item was “to what extent do you feel you receive emotional support from your friends?”. A higher mean score represented a higher level of received social support. Cronbach’s alpha was 0.86 in this study.

#### 2.2.6. COVID-19 Information Literacy

Three items were specifically developed to measure participants’ ability to understand, access, and distinguish correct COVID-19-related information (i.e., “I don’t understand the government’s information on COVID-19”, “I don’t know how to get COVID-19-related information”, and “I cannot tell which information about COVID-19 in the media/social media is true or wrong”). On a five-point scale (1 as not at all true, 5 as always true), a higher mean score from the item responses represented a lower COVID-19-related information literacy. Cronbach’s alpha was 0.72.

#### 2.2.7. COVID-19-Specific Worries

Two items were specifically developed to measure participants’ worries about the consequences of getting COVID-19 (i.e., “I worry that my employer will fire me if I get COVID-19” and “I am worried about getting COVID-19”). On a five-point scale (1 as not at all true, 5 as always true), a higher item score represented a higher level of COVID-19-specific worry. Similarly to other studies examining the COVID-19-specific worries with respect to mental health outcomes in general populations in China [30], Taiwan [42], and Hong Kong [4], different aspects of COVID-19-specific worries were indicated by single items, and were found to be associated with outcomes including anxiety and psychological well-being.

#### 2.2.8. Sociodemographic Variables

Participants were also asked to self-report their socio-demographic variables (e.g., age, educational level, years working in Hong Kong, monthly income, marital status, religious affiliation, whether they had children in their home country, etc.).

### 2.3. Data Analysis

Descriptive statistics were computed. The reliability coefficients of the scales were indicated by their corresponding Cronbach’s alphas. Multivariate logistic regressions were used to examine the associations between the independent variables and anxiety symptoms (binary-coded (yes versus no) for probable anxiety, based on the suggested cut-off point of GAD-7 ≥ 10). For independent variables (including both continuous variables and categorical variables), unadjusted odds ratios (ORs) were obtained by separately fitting each variable against probable anxiety (univariate analyses). Factors that were found to be related to anxiety symptoms (*p* ≤ 0.05) were then entered into a multivariate logistic regression model using the enter method. Univariate logistic regression analysis was used to assess how socio-demographic factors, work-related settings, job arrangement factors, coping resource factors, and COVID-19-specific worries were associated with probable anxiety. Multivariable logistic regression examines the contribution of each variable in distinguishing between groups (with or without probable anxiety) while controlling for the other variables in the model. The use of binary coding would better allow us to identify risk and protective factors of probable anxiety (GAD-7 ≥ 10). Given that the same cut-off point for probable anxiety (as in other COVID-19-related studies in Hong Kong and the United States) [4,31] was used, we believe that it would allow an easier comparison of the significant contributors of probable anxiety with other existing studies.

### 2.4. Sample Size Planning

Based on previous data from a meta-analysis about the prevalence of anxiety among the general populations during the COVID-19 pandemic [3], we estimated that 25% of FDHs in Hong Kong would report high levels of anxiety symptoms. To detect the smallest between-group difference in odds ratio of 1.50 for the key independent variables, a total sample size of at least 293 was required (power and significance level at 0.80 and 0.05, respectively) (G*Power 3.1.2, Heinrich-Heine-Universität Düsseldorf, Düsseldorf, Germany). With a sample size of 295, we should be able to detect the expected effect size with sufficient statistical power.

## 3. Results

### 3.1. Participants’ Characteristics

All participants (*n* = 295) were females. Most of the participants were aged between 31 and 50 (74.2%). Around half of them were married (56.3%), with over a half (58.3%) having an education level of college and above. Most of them reported having children in their home country (80.3%). On average, they had worked in Hong Kong for 6.2 years (SD = 6.4), with most of them (82.7%) reporting a monthly income lower than 5000 HKD (i.e., 645 USD). On a scale from 1 (beginners) to 10 (very proficient), the participants, on average, rated their level of English proficiency at 6.8 (SD = 2.6). Over 95% were affiliated with a religion and were social media users. During the COVID-19 pandemic, the participants worked 13.0 h (SD = 3.6) per day on average (Table 1).

### 3.2. Logistic Regression Analyses

Univariate logistic regression results showed that crowdedness in the household, insufficiency of preventive equipment, increased workload during COVID-19, COVID-19 information literacy, and COVID-19-specific worries were significantly associated with participants’ probable anxiety (GAD-7 ≥ 10). Specifically, those who lived in a household with higher crowdedness reported a greater risk for probable anxiety (OR = 1.31, 95%CI: 1.03, 1.67). Participants who reported higher levels of insufficiency of preventive equipment (OR = 1.54, 95%CI: 1.22, 1.95), workload during COVID-19 (OR = 1.95, 95%CI: 1.41, 2.71), worries about being fired if getting COVID-19 (OR = 1.43, 95%CI: 1.18, 1.73), and worries about getting COVID-19 (OR = 1.43, 95%CI: 1.09, 1.87) were associated with a greater risk for probable anxiety. Higher COVID-19 information literacy was associated with a lower risk for probable anxiety (OR = 0.67, 95%CI: 0.53, 0.87).

Our analyses found that there were no background variables showing significant associations with probable anxiety in univariate logistic regressions (Table 2); those variables were not adjusted for in the final multivariate regression model. The results from the multivariate logistic regression analysis indicated that after controlling for univariately significant factors in the model, insufficiency of preventive equipment (OR = 1.58, 95%CI: 1.18, 2.11), increased workload during COVID-19 (OR = 1.51, 95%CI: 1.02, 2.25), and worries about being fired if getting COVID-19 (OR = 1.32, 95%CI: 1.04, 1.68) were still significantly and positively associated with probable anxiety in the sample (Table 3)

## 4. Discussion

This was one of the first studies to examine the psychosocial correlates of probable anxiety among FDHs in Hong Kong amid the COVID-19 pandemic. Despite the relatively better-controlled pandemic situation in Hong Kong compared to other countries, 25% of FDHs in Hong Kong reported having probable anxiety (GAD-7 ≥ 10) during the pandemic. Using the same measurement scale, our sample reported a comparable prevalence of probable anxiety with the general populations in the United Kingdom [43] and in Bahrain [44], but a prevalence lower than those of the general populations in Hong Kong [4], China [45], and Japan [46]. Our findings suggested that we should not ignore the mental well-being among FDHs in Hong Kong. In addition, this study highlighted that insufficiency of protective equipment, increased workload due to COVID-19, lower COVID-19 information literacy, and higher worry of being fired if getting COVID-19 contributed to those helpers’ probable anxiety.

### 4.1. Working Household Environment, Job Arrangement, and Anxiety Symptoms

Crowdedness in working households and household size were associated with probable anxiety in the univariate analysis. However, those variables were no longer significant in the multivariate analysis after considering other variables including job arrangement, coping resources, and COVID-19 specific worries. Similarly, compared to other psychosocial variables, household size did not contribute significantly to anxiety symptoms among a general population in China [29]. Regarding job arrangements, insufficiency of protective equipment and increased workload due to COVID-19 were associated with higher anxiety symptoms among the FDHs. Among the significant factors in univariate analysis, job arrangement variables remained significant in the multivariate model. The findings suggested that insufficiency of protective equipment and increased workload during COVID-19 might be the strongest contributors to probable anxiety among the FDHs in Hong Kong. In occupational health literature, heavy workload (e.g., constant time pressure, more demanding jobs) has been found to be associated with poorer mental health among employees [37]. Recently, workplace responses to COVID-19 have also been found to be associated with psychological distress among Japanese workers [47]. The phenomenon applies to anxiety symptoms among FDHs in Hong Kong during the COVID-19 pandemic.

### 4.2. Social Support, COVID-19 Information Literacy, and Anxiety Symptoms

Contrary to our hypotheses, social support from employers, family, friends, and community organizations was not associated with probable anxiety in univariate analysis. Prior studies did not report consistent findings on the protective role of social support in mental health among individuals affected by COVID-19. For example, partner and peer social support were not associated with anxiety symptoms among US young adults [48], whereas social support was associated with lower anxiety symptoms among Chinese medical staff treating patients with COVID-19 [49] and college students in China [50]. Prior studies have suggested that other dimensions of social support (e.g., subjective evaluation about whether the social support that is received by a person is perceived by that person as valuable and effective) matter [24,51]. In rapidly changing pandemic situations like COVID-19, the specificity and usefulness of social support might be even more important. We suggest further studies to further elucidate how social support received and satisfaction with social support contribute to people’s well-being in pandemic situations.

Regarding informational resources, this study was unique in showing that lower literacy in COVID-19-related information was associated with probable anxiety in the univariate analysis. It is also noteworthy that most of the FDHs are social media users, but over 70% of them could not tell which information about COVID-19 in the (social) media is true or not. Support for obtaining and interpreting timely and accurate COVID-19 information will be important for those FDHs. However, COVID-19 information literacy did not show a significant association with probable anxiety in the multivariable analysis. Just having the ability to understand the information and distinguish true information, as well as knowing where to find information, might not be sufficient to reduce anxiety symptoms. A prior study has indicated that satisfaction with information was associated with people’s mental health among individuals in Bahrain amid the COVID-19 pandemic [44], implying that the quality of information might count. Further studies on how the quality of COVID-19-related information affects FDHs’ mental health are warranted.

### 4.3. COVID-19-Specific Worries and Anxiety Symptoms

This study was unique in revealing that COVID-19-specific worries about being fired if contracting COVID-19 were associated with anxiety symptoms among FDHs in Hong Kong. Previously, worries about the consequences of getting COVID-19 (e.g., maintaining employment during the subsequent economic downturn, getting treated for COVID-19 if contracting it) were associated with anxiety symptoms among the general population in China [30] and young adults in the US [31]. Our study highlighted that the worries about being fired if getting COVID-19 (rather than health-related worries about getting COVID-19) were even more salient in contributing to probable anxiety among FDHs. According to the immigration laws in Hong Kong, the helpers are obliged to find a new contract within two weeks or pay for a visa extension if they are unemployed [15]. As breadwinners for their families in their hometowns, FDHs tend to perceive a great deal of pressure to safeguard their jobs during the economic downturn in the COVID-19 pandemic. Such stressors might exacerbate the stress associated with the employment insecurity and contribute to anxiety symptoms [22].

### 4.4. Limitations

This study was subject to several limitations. First, it was a cross-sectional study, so the tested relationships were not causal. Future studies should examine how the changes in the independent variables could temporally predict anxiety symptoms using longitudinal designs. Second, we only recruited FDHs in Hong Kong at public places just after the loosening of social gathering restrictions in public places from a maximum of four people to eight people [34]. Our participants might represent more socially integrated individuals. The findings might only be partially applicable to foreign domestic helpers of other ethnicities (e.g., Indonesian) and varied social dynamics [52]. Third, we only recruited participants who could finish the questionnaire in English. Given that English is the language of written and spoken communication by FDHs employed in Hong Kong, this study used a similar research design to that of other local studies by administering the questionnaire in English [25,53]. We acknowledge that such inclusion criteria might compromise the representativeness of the sample; however, we believe that our strategy could still be acceptable in providing preliminary evidence for novel research areas. Future research may consider allowing participants to choose to answer the questionnaire in either their native language (e.g., Tagalog) or English. Fourth, our sample was quite highly educated, with more than half of the sample reported to have college education. However, local studies among Southeast Asian domestic helpers reported comparable education levels with our sample (over 90% received secondary and tertiary education) [25,54]. Generalizations of our findings to FDHs with different demographic characteristics must be made with caution. Fifth, this study only examined the roles of job arrangements (e.g., workload, sufficiency of protective equipment), coping resources (social support, COVID-19 information literacy), and COVID-19-specific worries in probable anxiety among FDHs in Hong Kong. Other factors may also be at play. For example, individual characteristics (e.g., ability to tolerate distress, resilience) have been found to be important contributors to people’s well-being during COVID-19 [31]. Considering these variables could allow a more comprehensive capture of the contributors to people’s mental health outcomes in response to the pandemic. Sixth, this study targeted a population that is underserved in the literature. Some measurement scales were newly developed for the context of COVID-19 that were not fully validated in this population, or single items were used to represent the constructs. The use of specifically developed single items as independent variables was also apparent in studies examining the psychosocial correlates of COVID-19-related mental health outcomes among the general populations in different countries and regions (including Hong Kong, Taiwan, China, and the US) [4,29,30,42]. Even with the limitations of single items as indicators of independent variables, we believe that those measures were still empirically acceptable for testing novel hypotheses in the context of the COVID-19 pandemic. In addition, the self-developed scales reported satisfactory psychometric properties. Replicating the study with different scales measuring the same concepts may help to further validate our findings.

### 4.5. Implications

Our findings suggested that increased workload, insufficiency of protective equipment, and worries of being fired if getting COVID-19 could be important determinants of anxiety among FDHs. Addressing those variables through public health amenities (e.g., specific hotlines/helplines that specifically serve FDHs, provision of protective equipment through community services, providing more information, and helping them to understand employee-related rights) might better equip them to cope with the pandemic. The International Labor Organization has also urged for social protection for those migrant workers in the COVID-19 pandemic. Our findings might highlight that policies covering a set of practical solutions for job arrangements should be in place [55]. Workers’ organizations could also help to support the promotion and protection of the FDHs’ rights during the pandemic.

It is important to note that mental health services specifically for FDHs are lacking in Hong Kong. Future studies should also examine COVID-19-related mental-help-seeking behaviors among migrant workers. It has been found that FDHs reported significantly higher intentions to seek help from family and friends than from doctors and other health professionals when feeling distressed [56]. We speculate that a similar pattern might be apparent in the context of COVID-19. Examining the factors associated with mental-help-seeking behaviors could help to further tailor interventions that could address the psychosocial needs among foreign domestic helpers.

## 5. Conclusions

In response to the COVID-19 pandemic, psychological health among FDHs in Hong Kong should not be ignored. This study therefore contributed by examining the important psychosocial correlates of probable anxiety in this underserved and vulnerable population. Having increased workload, insufficient protective equipment, and worries about being fired if getting COVID-19 uniquely contributed to probable anxiety among FDHs in Hong Kong amid the COVID-19 pandemic. Our findings might shed light on the importance of addressing employment-related rights among FDHs in Hong Kong during pandemic situations.

## Figures and Tables

**Table 1 ijerph-17-08102-t001:** Characteristics of the participants (*n* = 295).

Demographic Variables	Frequency (%)/Mean (SD)
Age (years)	
18–25	2 (0.7)
26–35	104 (35.3)
36–45	123 (41.7)
46–55	52 (17.6)
Over 55	11 (3.7)
Missing	3 (1.0)
Education level	
Primary/junior secondary school	16 (5.4)
Senior secondary/high school	106 (35.9)
College or above	172 (58.3)
Missing	1 (0.3)
Marital status	
Single	105 (35.6)
Married	166 (56.3)
Separated/Divorced/Widowed	21 (7.1)
Missing	3 (1.0)
Have children in home country	237 (80.3)
Have religious affiliation	284 (96.3)
Years of working in Hong Kong	6.2 (6.4)
<1 year	23 (7.8)
1–3 years	56 (19.0)
3–5 years	59 (20.0)
5–10 years	64 (21.7)
10–20 years	30 (10.2)
20 years or above	18 (6.1)
Missing	45 (15.3)
Monthly income (in HKD)	
Below 4630	88 (29.8)
4630–4999	156 (52.9)
5000–5499	23 (7.8)
5500 and above	16 (5.4)
Missing	12 (4.1)
Average number of working hours per day during COVID-19 outbreak	13.0 (3.6)
<8 h	14 (4.7)
8–13 h	109 (36.9)
13–20 h	141 (47.8)
>20 h	8 (2.7)
Missing	23 (7.8)
Being a social media user	283 (95.9)
District where they were currently living	
New territories	127 (43.1)
Kowloon	49 (16.6)
Hong Kong Island	108 (36.6)
Missing	11 (3.7)

**Table 2 ijerph-17-08102-t002:** Odds ratios and confidence intervals of background variables associated with probable anxiety (General Anxiety Disorder-7 (GAD-7) ≥ 10) (*n* = 295).

Demographic Variables	OR (95%CI)
Age (years)	
18–30	1
31–40	1.51(0.71, 3.18)
41–50	0.61(0.26, 1.43)
51 or above	0.62 (0.18, 2.16)
Education level	
Primary/junior secondary school	1
Senior secondary/high school	0.79 (0.23, 2.68)
College or above	1.16 (0.36, 3.78)
Marital status	
Married	1
Single/Divorced/Widowed/Separated	1.00 (0.58, 1.70)
Have children in home country	
no	1
yes	0.70 (0.35, 1.39)
Monthly income (in HKD)	
Below 4630	1
4630–4999	0.55 (0.31, 1.00)
5000–5499	0.60 (0.20, 1.77)
5500 and above	0.71 (0.21, 2.41)
Average working hours daily during COVID–19	
≤8 h	1
8–16 h	1.25 (0.51, 3.06)
>16 h	1.43 (0.44, 4.61)
Self–rated English level	1.04 (0.93, 1.16)
Being a social media user	
no	1
yes	0.84 (0.17, 4.12)

**Table 3 ijerph-17-08102-t003:** Odds ratios and confidence intervals of independent variables associated with probable anxiety (GAD-7 ≥ 10) (*n* = 295).

Variables	Mean (SD)	ORu ^ζ^ (95%CI)	Wald	df	*p*	ORm ^γ^ (95%CI)	Wald	df	*p*
Work environment variables									
Crowdedness in the household	3.04 (1.13)	1.31 (1.03, 1.67) *	4.84	1	0.03	1.03 (0.76,1.39)	0.03	1	0.86
Number of people in the household ^$^	4.51 (1.48)	0.97 (0.80, 1.16)	0.14	1	0.71	-			
Job arrangement variables									
Insufficiency of protective equipment	1.64 (1.05)	1.54 (1.22, 1.95) ***	13.12	1	0.00	1.58 (1.18, 2.11) **	9.27	1	0.002
Workload during COVID-19	2.45 (0.88)	1.95 (1.41, 2.71) ***	16.12	1	0.00	1.51 (1.02, 2.25) *	4.20	1	0.04
Coping resource variables									
Social support ^$^	3.42 (1.03)	1.21 (0.93, 1.57)	2.01	1	0.16	-			
COVID-19 information literacy	3.57 (1.07)	0.67 (0.53, 0.87) **	9.52	1	0.002	0.78 (0.58, 1.04)	2.83	1	0.09
COVID-19-specific worries									
Worries about being fired if getting COVID-19	3.42 (1.60)	1.43 (1.18, 1.73) ***	13.06	1	0.00	1.32 (1.04, 1.68) *	5.36	1	0.02
Worries about getting COVID-19	4.17 (1.24)	1.43 (1.09, 1.87) **	6.77	1	0.009	1.21 (0.87, 1.68)	1.21	1	0.27

Note: * *p* ≤ 0.05, ** *p* ≤ 0.01, *** *p* ≤ 0.001; **^ζ^**: Odds ratios not adjusted for background variables; **^γ^**: Odds ratios in the final multivariate logistic regression model; ^$^: Not entered into the final multivariate logistic regression model; Wald:

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
