# Peer review of "Feeling Anxious amid the COVID-19 Pandemic: Psychosocial Correlates of Anxiety Symptoms among Filipina Domestic Helpers in Hong Kong"

_ijerph, 2020, doi:10.3390/ijerph17218102_

Round 1

Reviewer 1 Report

This manuscript entitled "Feeling anxious amid the COVID-19 pandemic: Psychosocial correlates of anxiety symptoms among Filipina domestic helpers in Hong Kong" which aimed to  examine the psychosocial correlates of anxiety symptoms among FDH in Hong Kong amid the COVID-19 pandemic.

This study is quite interesting. However, some issues should be addressed:

1. Abstract:- Include more results, mainly statistical values.- Include a conclusion 

2. Introduction

  • If this manuscript will be accepted, the authors must update the following information in the first paragraph: " August 11, 2020, there have been 19,814,753 COVID-19 confirmed cases from 218 countries/regions". Moreover, for this sentence must be included a reference.
  • In general, the introduction section is very confuse for the reader. For instance, the authors have written in several subsections what they are proposing in their study. However, this is very repetitive as well as is writing in different way, as follow: "we focused on examining...", " We aimed to examine if social...", " this study attempted to examine if such",  " This study attempted to examine if these...", " This study examined the...". The authors must put efforts to improve the introduction section, specially the flow.

3. Methods

  • In the first sentence should be clearly stated the study design.

4. Results

  • Tables should be improved. For instance, Table 3 for the variable "Crowdedness in the household" there is a number 1 as reference and not for the other variables. It should be clear for the reader how was built the tables.
  • Tables: if there is any bold number, must be the reason in the legend.

Reviewer 2 Report

Dear authors,

Thank you very much for a clearly presented study.

Some suggestions to improve the manuscript:

1) In this sentence, I guess, the past tense should applied instead of future:

2.2.1. Anxiety symptoms.
The General Anxiety Disorder-7 (GAD-7) will be used to measure participant’s levels of anxiety
over the last 2 weeks [25].

2) Could you provide information on sex/gender characteristics of participants - if there were also men, % of women and men.

3) Also, if these workers have their own children or not, might contribute to the general workload and anxiety. If you have this information, please, provide. If not, include in the Study Limitations.

Reviewer 3 Report

This is an interesting study about an important and under-research section of  the employment market, that of domestic helpers. The context (Hong Kong) and sample group (Filipino respondents) also make a worthwhile contribution to the extant literature in the area. The study is generally well-written and clearly described. However, there are a number of aspects that, in my view, need attention before the article can be considered further for publication in IJERPH. These can be listed as follows.

  1. The authors have presented detailed descriptions of the measuring instruments and their internal reliability coefficients, barring that of COVID-19 'specific worries' ( page 5, section 2.2.7 where Chronbach alpha is not reported). However, there is very little detail on the construct and discriminatory validities of these scales used in the Hong Kong context and with Filipino respondents. This aspect needs some further explanation and elaboration. For example the scale measuring workload during COVID-19 ( page 4 section 2.2.3) had been used with German employees but not in China or with English speaking Filipinos. 
  2. The authors use logistic regression analysis, although the measuring instruments in the study use Likert-type scales which are normally considered appropriate for parametric statistical analysis.  Therefore, the authors need to explain and justify their use of logistic regression in their study.
  3. The study describes the effects of the COVID-19 pandemic on the mental health of Filipino domestic helpers, but there is no criterion variable that indicates the severity of the mental distress experience by these workers. How many had sought specialist healp for their mental health? In general terms, in the Hong Kong situation what proportion of  domestic helpers, whether Filipino or otherwise, had to seek medical, psychiatric or clinical psychological help since the outset of the COVID-19 pandemic?  
  4.  It follows from item 3 above that the practical implications of the study need to be strengthened by exploring the effects of domestic workers' mental health on  Hong Kong's public health's amenities, resources and competency to offer help.

   I wish you well in your revision of this interesting article

Reviewer 4 Report

Dear Authors,

thank you for the opportunity to review this study. It is a well-designed and interesting study about domestic helpers. It provides interesting data about how insufficiency of protective equipment or workload could increase the anxiety level. Psychological battery tests and data analysis are the strength of this study. Nevertheless, there are some minor issues, to which I would like the authors to refer. Also, it seems that the article could use a professional proofreading service because I found some minor language inconsistencies. I suggest publishing this article after the minor revision.

Minor issues:

The last author name is written in lower case (Tak-fai Lau)

2.2.1 Anxiety symptoms. – citation [2] does not refer to the validation study.

3.1 Put a dot after the paragraph.

Under the table 3, you refer to p as a  ‘p ≤ .01’ with omitted 0. In 2.3. Analytic plan (which I suggest to rename to data analysis) you write (p≤0.01). Please stick to one way of referring, including spaces.

4.1. ‘…the strongest contributors of the anxiety…’ – correct preposition

4.1 ‘…COVID-19 have also found to..’ correct to ‘…COVID-19 have also been found to…’

4.2 ‘…information literacy did not show significant…’ put the article before significant

4.3 ‘…worries about being fired if getting COVID-19 (rather than health-related worries about getting COVID-19) was even…’ – worries were

Correct study contributions according to journal requirements

Correct the reference section. In some places, references are disorganized and chaotic. To reference 1 put the website link, in the websites reference journal require to write the full date (accessed on Day Month Year). In reference 4 there is a lot of acronyms, correct capital letters, etc. Check all references and correct them.

Reviewer 5 Report

This study tackles a relevant and timely issue, i.e. psychosocial predictors of anxiety symptoms among migrant workers amid the COVID-19 pandemic. In general, I have no serious objection to this manuscript being published, although there are some points I think should be addressed by authors (see below).

  • I feel like the introduction would benefit from the integration of more general reasoning and literature, not directly related to the COVID-19 crisis, in particular in order to provide a rationale for the choice of using anxiety as outcome, and not e.g. measures of emotional well-being or other symptoms, as well as for the choice of predictors.

  • English needs revision. Some examples:

    • the first two lines of paragraph 1.2 read “Given that COVID-19 is highly infectious, the working household environment in could be particularly important to FDH’s well-being”, I think there is a typo;

    • the last two lines of paragraph 1.2 read “Therefore, we expected that increased workload during COVID-19 and insufficiency of protective equipment were associated with anxiety symptoms among FDH in Hong Kong”. I would change “were associated” to “would be associated”. The same goes for paragraph 1.5 (“… were associated with higher/lower anxiety symptoms” should probably be changed to something like “are expected to be associated with…”).

    • Paragraph 1.3 refers to the relationship between social support, information and anxiety, but the direction (positive or negative) of the associations found in literature is not explicitly stated.

  • I understand the need of implementing new scales to measure specific aspects of the COVID-19 crisis, but I find problematic the choice of using single items, for which no reliability measure is available (regarding working environment and COVID-19 specific worries) for some crucial aspects of the statistical analysis. I think this choice should be justified if possible, or at least mentioned as a serious limitation.

  • The choice of binary coding should be justified as well. Why did authors not adopt the anxiety score as an outcome? Is this due to statistical considerations (e.g. distributional properties of the variable) or to an a-priori choice?

  • I find the presentation of participant characteristics confusing, as some information is reported in paragraph 2.1 (Participants and recruitment strategies) and some in 3.1 (Participant characteristics), while the sample size is only reported in 2.4 (Sample size planning).

  • Only odds ratios are reported for the logistic regressions. I suggest to also report effect sizes, degrees of freedom, the Wald statistic and exact p values. Also, authors refer to the portion of variance explained by the model (in the limitations section, 4.4) but do not provide any R squared value in the results section.

Round 2

Reviewer 1 Report

All suggestions were addressed by the authors.

Good luck with this publication.

Author Response

Thank you for your comments! We have proofread the manuscript again and ensured the style is in line with the journal requirement.

Reviewer 3 Report

Thank you for dealing with the various problems raised in my report. I believe you have handled these satisfactorily, barring a brief explanation in the text itself of why you log logistic regression as a statistical methodology. Once this brief addition is made to the existing text, I believe your paper is ready for publication in IJERPH.  
